# The Bottle House: Upcycling Plastic Bottles to Improve the Thermal Performance of Low-Cost Homes

Nwakaego C. Onyenokporo [1],*, Arash Beizaee [2], Olutola F. Adekeye [3] and Muyiwa A. Oyinlola [4],*

1   Leicester School of Architecture, De Montfort University, Leicester LE1 9BH, UK
2   School of Architecture, Building & Civil Engineering, Loughborough University, Loughborough LE11 3TU, UK; a.beizaee@lboro.ac.uk
3   Department of Architecture and Built Environment, University of Nottingham, Nottingham NG7 2RD, UK; olutola.adekeye@nottingham.ac.uk
4   Institute of Energy and Sustainable Development, De Montfort University, Leicester LE1 9BH, UK
*   Correspondence: nwakaego.onyenokporo@my365.dmu.ac.uk (N.C.O.); muyiwa.oyinlola@dmu.ac.uk (M.A.O.)

**Abstract:** Due to the effects of climate change, diminishing natural resources, and continuous urbanization, there is an increasing need for buildings to be more sustainable. This study explores the potential of upcycling plastic waste for the sustainable construction of low-cost homes in developing countries and contributes to filling the gap in existing studies regarding qualitative results of the in situ performance of buildings made from upcycled materials. This study compares the Bottle house with conventional buildings made of mud and cement. This study seeks to encourage the adoption of the bottle house concept for affordable housing by conducting a thermal comfort survey of its occupants. To obtain the thermal sensation vote (TSV) of the occupants, thermal comfort questionnaires were developed based on the seven-point ASHRAE thermal sensation scale. Additionally, a Testo 480 multifunction meter, which comprised an anemometer, radiant globe thermometer, air thermometer, and relative humidity probe, was used to calculate the predicted mean vote (PMV) concurrently. From the results of the TSV, mean votes of the participants of −2.0, 2.0, and 2.4 were observed for the bottle house, mud houses, and cement houses, respectively. In comparison, adjusted PMV mean values of 1.9, 2.1 and 2.1 were recorded for the bottle house, mud houses, and cement houses, respectively. The TSV and PMV results both indicate that the occupants of the bottle house felt more thermally comfortable when compared to occupants in the other dwellings. This can be attributed to the measures incorporated during the construction of the bottle house. Furthermore, the use of a simulation study helped proffer solutions to further improve the indoor temperatures of the buildings used in this study. The results of this paper will provide evidence of the prospects of upcycling plastic waste for construction and its impact on occupant's thermal comfort when compared to conventional building materials.

**Keywords:** sustainable building materials; upcycled waste; plastic waste; thermal comfort; circular economy; thermal sensation vote (TSV); predicated mean vote (PMV)

## 1. Introduction

All over the world, there is a growing need for buildings to be more sustainable due to climate change, diminishing natural resources, and continuous urbanization. Low- and middle-income countries (LMICs) or developing countries have been reported to be most affected by the effects of climate change, especially with increasing urbanization and economic development [1]. The need to produce more buildings to meet housing requirements for the growing populations of these countries will also lead to the further depletion of the natural environment. Due to the continuous utilization of resources, there is also a global problem concerning waste disposal. In 2016, the global waste generated stood at 2.01 billion tons [2], and according to the World Bank [3], this figure is projected to

increase by 70% in 2050 due to increasing population and urbanization. One of the major wastes being disposed of is plastic waste, with about 353 million tons produced in 2019, which is projected to increase to 1014 million tons by 2060 based on the trendline [4,5]. According to Adefila et al. [6], developed countries typically have infrastructure or policies targeted at the reuse or recycling of waste. However, infrastructure and policies for waste management are still underdeveloped in LMICs. In addition, compared to high-income or developed countries, only 4% of waste generated in low-income countries is recycled [3].

Using upcycled materials for building low-cost houses is generally considered a viable solution to address inadequate housing in low-income communities [7]. Several scholars have reported using upcycled materials such as plastic waste [8–10], agricultural waste [11–13] and fiber waste [14–16]. This field of study is important because plastics are increasingly imported in large quantities into many countries with limited infrastructure for sustainably managing large volumes of plastic waste. Furthermore, Tarabieh et al. [17] agree that since the recycling of plastic waste in developing countries is still low, reuse or upcycling should be advocated for as viable solutions. Similarly, Ahmed [18] recommends the reuse of plastic waste to produce affordable and eco-friendly construction materials. The reuse of plastic waste for use in buildings is also reported to have a lower cost than conventional materials [17]. This study, similarly, advocates for the reuse of these plastic wastes, usually disposed of indiscriminately, for the sustainable construction of dwellings, especially in LMICs, as this will provide a partial solution to addressing current housing deficits in these countries and provide alternative building materials for the low-cost and sustainable construction of dwellings in these countries. This will also serve to promote a circular economy. Adopting a circular economy through the upcycling of landfilled waste materials can reduce the cost of buildings, especially in low- or middle-income countries, as the major or cost-intensive portion of construction is the cost of materials [6]. In addition, the upcycling of these wastes should be encouraged as it will serve to encourage sustainable communities and the adoption of sustainable waste management practices in these regions or countries in accordance with UN Sustainable Development Goal 11.

Although research on plastic bottle reuse is not new, as evidenced by the existing literature, the focus has been on the strength of bottle composites, and not much research has focused on the thermal performance of plastic bottle composites when used in construction until recently [17]. The comprehensive review presented by [19] shows that numerous studies have investigated the thermal performance of upcycled materials for constructing homes; however, these studies have mainly focused on the component level. A few studies [17,20] have predicted the performance of buildings from upcycled materials using simulation and quantitative studies to determine the effect of overheating in such buildings [21]. Tarabieh et al. [17] evaluated the thermal performance of sand-filled plastic bottles using a simulation study and compared it to traditional composite brick walls in Egypt. In their study, they compared three brick wall typologies (one with plaster, 200 mm brick, insulation, and gypsum board, and the other two with plaster and bricks of two different thicknesses) with two variations in the bottle bricks samples (1.5 L and 0.75 L). They observed that the bottle block samples made with 1.5 L bottles had lower thermal transmittance coefficients (u-values) than the brick wall sample made from 100 mm brick and plaster, although it was higher than the u-values of the 200 mm brick wall samples made with plaster, insulation, and plasterboard. Similarly, Kougnigan et al. [22] investigated the thermal performance of concrete blocks containing plastic bottles and crushed clay. They recorded a decrease of up to 50% in the thermal conductivity of blocks containing plastic bottles when compared to the control sample. Nevertheless, a study conducted by Mokhtar et al. [23] involved in situ thermal measurements of an eco-house built with plastic bottle bricks. They produced bricks using 250 mL and 1.5 L plastic bottles to compare with common clay bricks in Malaysia. They constructed a bottle eco house prototype in order to evaluate its thermal performance and influence on the indoor temperature and compared it with a standard brick house. They recorded minimum daily temperatures of 28.6 °C and 29.7 °C for the brick house and plastic bottle eco house, respectively. Although

they observed that the indoor temperatures for both house types were in a similar range, even with limited ventilation in the bottle house; however, they recorded a maximum temperature of 33.8 °C for the bottle house, which is lower than that of the brick house, which was 34.1 °C. These studies lend evidence that plastic bottle walls can provide better or comparable thermal performance when compared to conventional brick walls.

As evidenced by the aforementioned studies, existing research on the thermal performance of these plastic bottles in construction has focused on quantitative measurements, such as thermal conductivity and thermal transmittance coefficient measurements, which are important when considering the flow of heat into or out of the building [17,19,22,23]. However, there is a dearth of research data on experimental in situ measurements and/or qualitative studies. Only one qualitative study has been conducted on plastic bottle houses [24]. However, the focus of their study was to garner the perception of the construction team at the NGO Developmental Association for Renewable Energies (DARE) in Yelwa village, Nigeria, on the design and construction of their bottle house prototypes.

This paper, therefore, contributes to filling this significant knowledge gap by reporting on the in situ thermal performance of a low-cost building made from upcycled plastic bottle waste while also focusing on garnering qualitative data from the building occupants. Furthermore, this paper complements the few existing studies that have explored thermal comfort in low-income dwellings in the developing world, such as [25], who studied thermal comfort in low- and middle-income dwellings in Abuja, Nigeria, and [26] who conducted a similar study in Uganda. Notably, this study also makes a unique contribution to the existing research as it compares the performance of a bottle house (which has an unconventional material as the envelope) with conventional buildings (mud and cement).

There still exists very low adoption of upcycled plastic bottles in real-life projects, although it has been used in many cases for the construction of low-cost and eco-friendly homes [24]. As this is a new building made using upcycled plastic bottles, no data are available on the post-occupancy thermal sensation of occupants in this bottle house. This study seeks to encourage the adoption of the bottle house concept for affordable housing by conducting a thermal comfort survey of the occupants, as thermal comfort is an important parameter to consider for any building, especially regarding building energy performance. According to Richards and Hemphill [27], a qualitative study enables in-depth information to be obtained from the participants about their lived experiences. This, therefore, lends novelty to this present study, as the authors succeeded in garnering qualitative responses from the occupants of the bottle house and comparing this to data from other existing building typologies. Garnering the perception of end-users is also in line with social sustainability. As the participants were consulted before, during, and after the construction of the bottle house, it is also important to obtain post-occupancy data concerning their thermal comfort. Furthermore, this paper compares the results obtained to typical building typologies found in the study area. Post-occupancy evaluations to monitor indoor environmental conditions and assess occupants' thermal comfort using surveys have similarly been carried out by [25,28,29] using PMV and TSV thermal comfort models. BS EN ISO 7730:2005 [30] describes the predicted mean vote (PMV) as an index that predicts the mean value of the votes of a large group of persons exposed to the same environment on a seven-point thermal sensation scale (Table 1) based on the heat balance of the human body [31].

**Table 1.** Correlation between ASHRAE short-term thermal comfort scale (TSV) and PMV (adapted from [31]).

| TSV | PMV |
|---|---|
| Hot | +3 |
| Warm | +2 |
| Slightly warm | +1 |
| Neutral | 0 |

**Table 1.** *Cont.*

| TSV | PMV |
|---|---|
| Slightly cool | −1 |
| Cool | −2 |
| Cold | −3 |

The PMV thermal comfort model has been used extensively over the years in various studies and various climates. According to Fanger and Toftum [32], the PMV model predicts occupants' thermal sensations based on their level of activity, clothing, and environmental factors, such as air temperature, mean radiant temperature, air velocity, and humidity. It has the advantage of including the major factors affecting occupants' thermal sensation regardless of the HVAC system, indoor thermal environment, activity, or clothing level. However, Sharifi et al. [29] explain that PMV overestimates thermal sensations in warm climates without considering human acclimatization. PMV also has the delimitation of not being able to evaluate occupants' perception, hence the need for thermal sensation vote (TSV), as utilized in this study. Likewise, the PMV scale is the only scale correlated to TSV that can convert the verbal ranges to numerical values. According to Sharifi et al. [29], TSV allows for the evaluation of occupants' thermal sensation and is based on the individual's thermal perception. Another major reason for the discrepancy between TSV and PMV is that it is difficult to measure accurate values for metabolic rates and clothing insulation [33].

## 2. Innovative Design of the Bottle House

This study is part of the Bottle House project, which is an international, transdisciplinary collaboration between academia, industry, and end-users in a low-income community in Nigeria [7,34,35]. The project adopted a user-centered design approach for the design and construction of an affordable, sustainable home from upcycled materials; the walls were constructed using plastic bottles, the ceiling from used bamboo scaffolding, and the floor was created from recycled tiles. The Bottle House, shown in Figure 1, is situated in Abuja, the Federal Capital Territory of Nigeria.

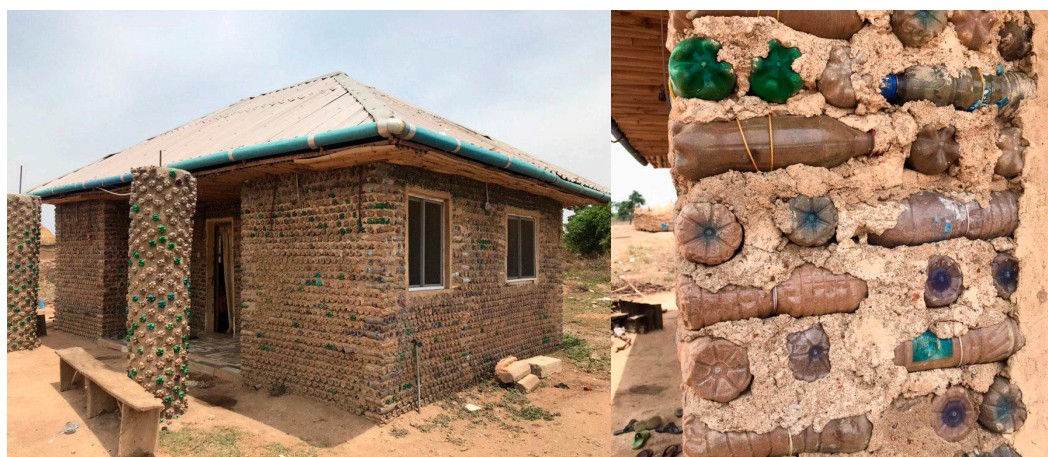

**Figure 1.** Bottle house in Paipe, Abuja, Nigeria.

The design framework adopted for the bottle house project was iterative and involved both designers and end users working collaboratively from the beginning to the completion of the design project. A diverse range of qualitative and quantitative research methods, such as observational study, experimental studies, focus groups, interviews, etc., were adopted throughout the project. Figure 2 provides a summary of the design framework, showing the different stages of data collection and analysis involved in the construction of the bottle house prototype.

| PLANNING AND DESIGNING | | | |
|---|---|---|---|
| Conduct a literature review to determine possible waste materials for upcycling | Observational studies to learn about the study context and determination of appropriate local waste materials for upcycling | Interviews and Focus groups with relevant stakeholders in the community to brainstorm potential design | Formulate prototype design using a transdisciplinary approach |
| TESTING AND OPTIMIZATION | | | |
| Conduct initial experiments on building components incorporating different local waste materials to determine viability | Present the design to the community to obtain feedback | Revise and deliberate on the final design of the prototype | Conduct final experiments on building components to optimize their design |
| IMPLEMENTATION AND CONSTRUCTION | | | |
| Enjoin stakeholders/community in the construction process via workshops | Collection and processing of plastic waste materials by researchers and participants | Begin construction of the Bottle house prototype | Interviews and Focus groups with community and relevant stakeholders to obtain feedback and evaluate user acceptability |
| POST-OCCUPANCY | | | |
| Obtain post-occupancy feedback from residents | | Conduct monitored temperature measurements inside the Bottle house for quantitative thermal analysis | |

**Figure 2.** Stages of the design framework integrating a user-centered design approach (adapted from [7,21]).

Following the design and construction of the bottle house prototype, a study on the post-occupancy thermal comfort of the occupants in the dwelling was conducted to determine their thermal perception. This is significant in closing the loop in the user-centered approach adopted in this study. Thus, this paper is significant as it compares the predicted thermal comfort performance to the actual thermal comfort experienced by the occupants.

### 3. Materials and Methods

*3.1. Climatic Conditions of the Study context*

The bottle house used in this study is located in Abuja, Nigeria. Abuja is situated in central Nigeria at a latitude of 9°07′ N and a longitude of 7°48′ E, with an elevation of 840 m (2760 ft) above the sea-level. The study area is located within the Savannah zone vegetation of the West African subregion with patches of rainforest [36,37]. Abuja's distinctive geographical features, such as the high altitudes and undulating terrain, act as a moderating influence on the weather of the city. According to Peel et al. [38], the Koppen climate classification subtype for Abuja is the tropical savanna climate (Aw). The average annual temperature is 26.0 °C in Abuja, with an annual precipitation of about 1469 mm [39]. The temperature ranges between 12° C and 40 °C from November to March, which is considered the Harmattan season (dry and dusty) [25,36,37]. Precipitation ranges from 305 to 762 mm (12–30 in.) in the rainy season (April–October) [25,36,37]. Data for both the comfort survey and monitored readings were collected in April, which is the end of the dry season and the beginning of the rainy season in Nigeria. It is also one of the months with the hottest temperatures throughout the year (Figure 3). The average outdoor temperature in April is 28.6 °C with a maximum of 33.8 °C and a minimum of 24 °C [39], while the relative humidity is 57% [37].

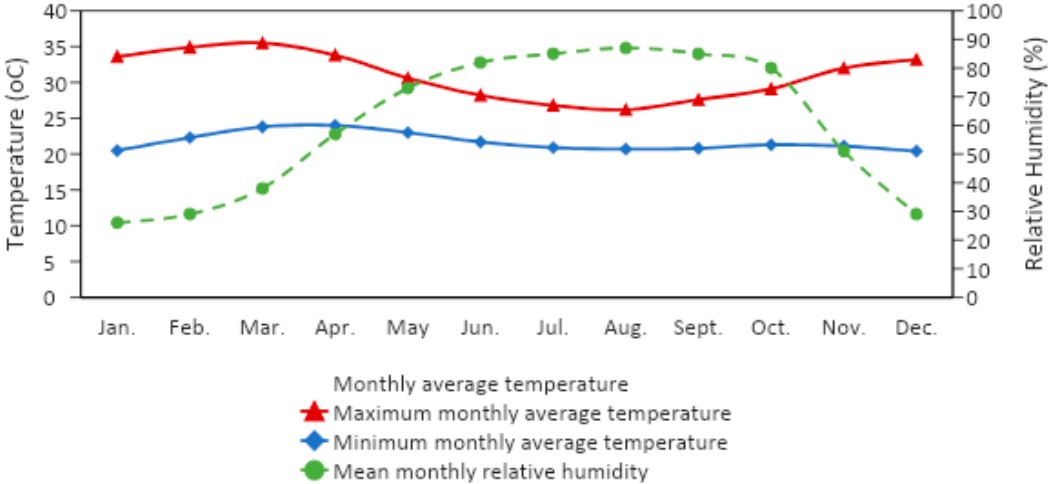

**Figure 3.** Average monthly temperature and relative humidity for Abuja (Weather data obtained from [39]).

*3.2. Description of Surveyed Dwellings*

For this study, a total of five buildings were used for the data collection. Two houses were built with cement-based masonry blocks, two houses were built with hand-formed mud, and one was the bottle house (Figure 4). According to the National Bureau of Statistics [40], the most common walling materials used for dwellings in Abuja include cement/concrete (78.9%), mud (14.5%), and bricks (6.4%). Therefore, this led to the selection of cement masonry houses and mud houses as used in this study. In addition, these were all selected from the same local community as the bottle house in Paipe, Abuja.

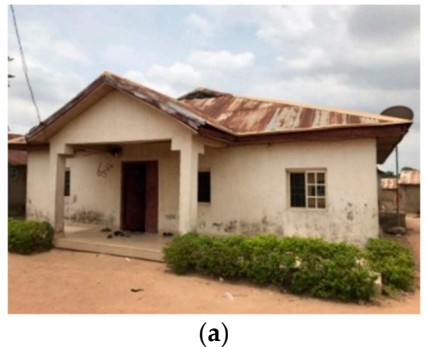
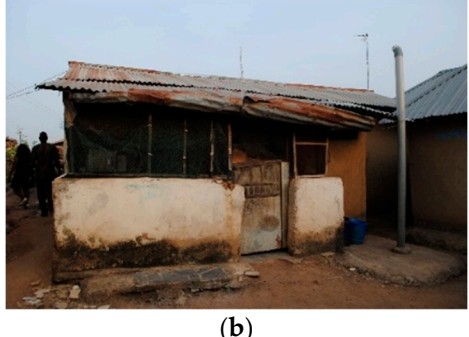
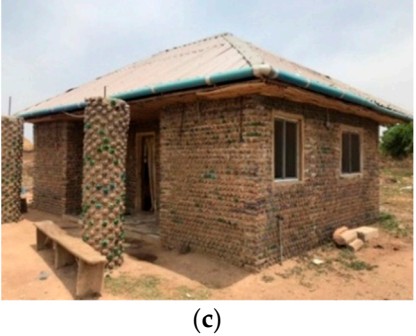

(**a**)  (**b**)  (**c**)

**Figure 4.** Buildings used for data collection: (**a**) cement, (**b**) mud, and (**c**) plastic bottle house.

The thermal transmittance (u-values) of the external walls of the bottle house, mud houses, and cement houses were 2.94, 2.62, and 4.0 W/m$^2$ K, respectively [21]. The bottle house (BH) was built using waste Polyethylene Terephthalate (PET) bottles collected within the local community, which were then filled with sand and water. The ceiling was made from sliced bamboo sticks. The external walls were constructed with a thickness of 225 mm to compare with the thickness of the typical masonry units used in the context of the study. The 225 mm external walls were composed of plastic bottles filled with sand, while bottles filled with water were used at the lintel level and also for the internal walls. The roof was constructed using Aluzinc sheeting, which is cold-rolled galvanized steel with a metal coating composed of aluminum (55%), zinc (43.4%), and silicon (1.6%) [7], while the ceiling was made from bamboo, which has a low thermal transmittance coefficient and served to reduce heat gains to the interior spaces from the roof. The floor was constructed with concrete and finished with reclaimed ceramic tiles. Figure 4 shows the similarities and differences between the selected buildings.

The cement-based house (CH) was constructed using 225 mm hollow sandcrete blocks, which are a conventional walling material used in Nigeria and stipulated within the National Building Code. They are typically produced using cement and sand without any coarse aggregate [41]. The walls were rendered with cement and sand plaster. The mud house (MH) was constructed using 200 mm hand-formed mud with a wooden framework. This was then rendered using cement and sand, similar to the cement-based house. The roofs of both house types are made from Aluzinc sheeting, and their ceilings are made of PVC strips, which are commonly used in the area due to their affordability. The floors are made of concrete and finished with ceramic tiles, similar to the bottle house.

*3.3. Qualitative Thermal Measurement Using Thermal Comfort Survey (TSV)*

A total of five families participated in this study, including the bottle house occupants. Two were from dwellings constructed with cement-based masonry blocks (also known as sandcrete blocks in Nigeria and its environs), and the other two were from dwellings constructed using hand-formed mud. The participants in the bottle house consisted of two adults and three children, while the mud houses had a total of five adults and one child, and the cement-based houses consisted of six adults altogether. It is important to note that for this study, only the responses from the adults were utilized in order to provide reliable and comparable data.

Thermal comfort questionnaires were developed based on the BS EN ISO 7730 standard [30] using the seven-point ASHRAE thermal sensation scale (Appendix A). These were administered to the occupants of the cement house, mud house, and bottle house, with three–four adult respondents from each house. These data were used to estimate the thermal sensation vote (TSV) and compare it to the calculated predicted mean vote (PMV). The participants were asked to complete the questionnaires during the physical measurement period. The researchers first discussed the study with the participants and obtained consent before collecting background information, which included the participants' names, ages, genders, etc. (Figure 5). Each participant completed the thermal comfort questionnaire in their living room at 15 min intervals. The survey also included the time at which the survey was completed.



**Figure 5.** Experimental design of thermal comfort survey.

The questionnaire evaluated occupants' thermal sensation, thermal preference, and acceptability, as seen in Tables 2 and 3. The survey was conducted when the participants had been seated in the living room for at least 15 min (Figure 6).

**Table 2.** Thermal sensation scale [Adapted from [30,42]].

| Scale | Thermal Sensation | Thermal Comfort | Thermal Preference | Thermal Acceptability | Personal Tolerance |
|---|---|---|---|---|---|
| +4 | | | | Clearly unacceptable | Unbearable/Intolerable |
| +3 | Hot | Very uncomfortable | Much warmer | Just unacceptable | Very difficult to bear/tolerate |
| +2 | Warm | Uncomfortable | Warmer | Just acceptable | Fairly difficult to bear/tolerate |
| +1 | Slightly warm | Slightly uncomfortable | Slightly warmer | Clearly acceptable | Slightly difficult to bear/tolerate |
| 0 | Neutral | Comfortable | Without change | | Perfectly bearable/tolerable |
| −1 | Slightly cool | | Slightly cooler | | |
| −2 | Cool | | Cooler | | |
| −3 | Cold | | Much cooler | | |

**Table 3.** Participant information.

|       | Sample Size | Gender       | Metabolic Rate (Met) | Clothing Insulation (Clo) |
|-------|-------------|--------------|----------------------|---------------------------|
| BH    | 3           | M = 1, F = 2 | 1.0                  | M = 0.36, F = 0.27        |
| MH 1  | 3           | M = 0, F = 3 | 1.0                  | M = 0.36, F = 0.27        |
| MH 2  | 4           | M = 3, F = 1 | 1.0                  | M = 0.36, F = 0.27        |
| CH 1  | 4           | M = 4, F = 0 | 1.0                  | M = 0.36, F = 0.27        |
| CH 2  | 2           | M = 1, F = 1 | 1.0                  | M = 0.36, F = 0.27        |

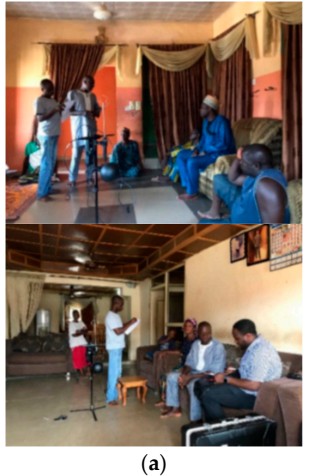 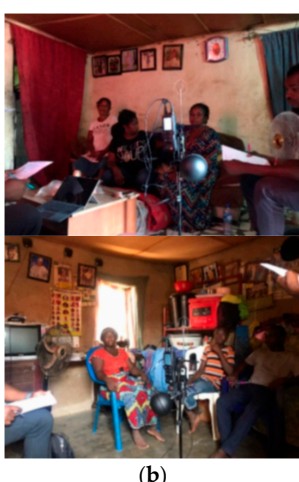 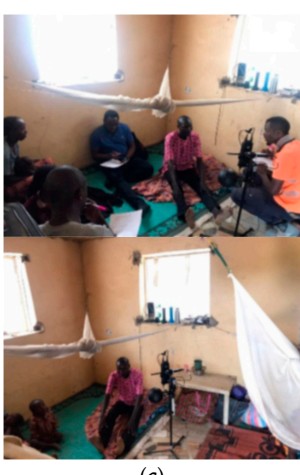

(**a**)          (**b**)          (**c**)

**Figure 6.** Data collection from occupants: (**a**) cement house, (**b**) mud house, and (**c**) bottle house.

*3.4. Quantitative Thermal Measurement Using Predicted Mean Vote (PMV)*

Quantitative thermal measurements were recorded in the living spaces, as seen in Figure 6. This was performed concurrently with the qualitative measurements for the purpose of comparison. The participants were observed by the authors in order to determine their corresponding metabolic rate (met) and clothing insulation level (clo), and the figures were calculated in accordance with BS EN ISO 7730. The predicted mean vote (PMV) estimates the mean response of occupants to their thermal environment and was determined using six parameters: air temperature, mean radiant temperature, relative humidity, air velocity, clothing insulation level, and metabolic rate. The thermal sensation vote (TSV), on the other hand, is the actual thermal sensation vote of the occupants and is obtained from the results of the thermal comfort survey by asking the occupants to vote on how they felt in relation to the ASHRAE seven-point thermal sensation scale [29]. During the field investigation, each participant took part in a single experimental session in their living rooms. For all the dwellings, the same experimental layout was used. During each session, measurements were taken both subjectively (using the ASHRAE 7-point thermal comfort questionnaire [31]) and objectively (using a Testo 480 multifunction meter [43] that included an anemometer, radiant globe thermometer, air thermometer, and RH). To control the metabolic rate, the participants were invited to sit down in their living room and either undertake sedentary work or rest during the sessions. The manufacturer's specifications of the Testo 480 multifunction meter [43] are presented in Table 4 below.

**Table 4.** Instruments for physical measurements.

| Name                   | Parameters               | Measurement Range | Accuracy                        |
|------------------------|--------------------------|-------------------|---------------------------------|
| Testo 480              | Air temperature          | 0 to +50 °C       | ±0.5 °C                         |
| multifunction meter    | Mean radiant temperature | 0 to +120 °C      | Class 1                         |
|                        | Relative humidity        | 0 to 100% RH      | ±(1.8% RH + 0.7% of m.v.)       |
|                        | Air velocity             | 0 to +5 m/s       | ±(0.03 m/s +4% of m.v.)         |

It is noteworthy to mention that all the houses were naturally ventilated, with some houses having mechanical fans to improve indoor comfort, although these were not in operation during the measurement period.

### 3.5. Dynamic Thermal Simulation Using DesignBuilder

The use of the EnergyPlus simulation engine in DesignBuilder was adopted within this study for comparison with the quantitative thermal measurements and to evaluate the effects of retrofit interventions on the selected buildings in order to recommend actions that would improve the thermal comfort of the occupants within the residential buildings in similar contexts. Full details are provided in Section 4.3. Table 5 shows the building specifications used for the simulation study.

**Table 5.** Building specifications used in DesignBuilder for modeling and dynamic simulation study.

| Parameters | Value |
|---|---|
| Location | Abuja, Nigeria |
| Latitude and longitude | 9.07°, 7.49° |
| Elevation above sea level | 360 m |
| Exposure to wind | Sheltered (to account for other buildings around the case study building) |
| Site orientation | 45° |
| Simulation weather data | Abuja, Nigeria (EPW hourly weather data file obtained from White Box Technologies) |
| Summer design weather data | For cooling design simulation for custom day Values obtained from measured outdoor temperature readings from site @30 min interval (maximum—35.3 °C, average—30.8 °C, minimum—28 °C) |
| Activity template | Residential dwelling with kitchen (zones mapped as common circulation area, kitchen, bedroom, or bathroom) |
| Walls | 225 mm Hollow sandcrete blocks * rendered with 15 mm cement plaster on both sides (U-value = 3.808 W/m$^2$ K) |
| Floor | Ceramic/porcelain tiles on 50 mm mortar and 150 mm cast concrete (U-value = 2.387 W/m$^2$ K) |
| Roof | Aluzinc roofing sheets ** on timber purlins and rafters with an air gap (U-value = 1.948 W/m$^2$ K) |
| Doors | Internal—wood; External—steel |
| Windows | Aluminum casement; Glazing—Single blue tinted 6 mm |
| Lighting template | Incandescent bulbs with lighting control |
| HVAC template | Natural ventilation-No heating/cooling (outside air was reduced from 5 ac/h to 2.5 ac/h due to the use of security bars and mosquito netting on windows. ***) |

* thermal conductivity derived from [44] ** [21] *** [45].

## 4. Results and Discussion

### 4.1. Thermal Comfort Survey (TSV)

The TSV values were calculated based on the values recorded in the questionnaires and are presented in Table 6. These responses were obtained from all participants in the afternoon. As this was carried out inside their homes, the clothing worn by the participants was casual and also lightweight, typical of hot climate regions. The women were dressed in a light blouse/t-shirt and a skirt/native wrapper, while the men typically wore t-shirts and trousers or a native kaftan typical of the local region. The average clothing insulation (clo) was 0.32, and the metabolic rate (met) was 1.0 (seated and relaxed), as indicated by

the majority of the participants. These values were calculated using the BS EN ISO 7730 standard [28]. This shows that their clothing and level of activity did not exaggerate their thermal perception [29,30].

**Table 6.** TSENS votes of building occupants.

| Participant | Q1: Your Activity Level Now | Q2: How Are You Feeling at This Precise Moment? | Q3: Do You Find This …? | Q4: Please State How You Would Prefer to Be Now? | Q5: How Do You Judge This Environment (Local Climate) on a Personal Level? | Q6: Please State Your Personal Tolerance of This Environment. Is It … | Q7: Time of Completing the Survey | |
|---|---|---|---|---|---|---|---|---|
| P1 | 2 | −2 | 2 | −3 | 1 | 0 | 12:10 | |
| P2 | 2 | −2 | 0 | 0 | 1 | 0 | 12:12 | BH |
| P3 | 2 | −2 | 0 | 0 | 1 | 0 | 12:12 | |
| P1 | 1 | 0 | 1 | −3 | 3 | 2 | 12:30 | |
| P2 | 2 | 1 | 0 | −2 | 1 | 0 | 12:30 | MH 1 |
| P3 | 2 | 3 | 3 | −3 | 4 | 3 | 12:30 | |
| P1 | 2 | 2 | 2 | −3 | 3 | 1 | 13:00 | |
| P2 | 2 | 2 | 2 | −2 | 3 | 1 | 13:00 | MH 2 |
| P3 | 2 | 3 | 3 | −3 | 3 | 3 | 13:00 | |
| P4 | 3 | 1 | 1 | −2 | 1 | 1 | 13:00 | |
| P1 | 2 | 2 | 0 | 0 | 1 | 0 | 11:42 | |
| P2 | 2 | 3 | 1 | −3 | 3 | 0 | 11:42 | CH 1 |
| P3 | 2 | 2 | 0 | −1 | 1 | 0 | 11:42 | |
| P4 | 2 | 2 | 0 | −1 | 1 | 0 | 11:42 | |
| P1 | 2 | 2 | 0 | −2 | 2 | 3 | 12:00 | |
| P2 | 2 | 3 | 0 | −2 | 3 | 0 | 12:00 | CH 2 |

All participants from the bottle house indicated a TSV of −2 (cool), while responses from participants in other house types ranged from 0 to 3 (neutral to hot), with those in the cement houses (CH 1 and 2) indicating only +2 and +3 (warm; hot). When asked how they would prefer to feel, most of the participants in the bottle house (BH) indicated that they did not want any change, while one occupant mentioned they would like to feel much cooler. In contrast, the majority of the remaining 13 participants indicated that they would rather be cooler or much cooler. This disparity is also observed in the TSENS votes for Q5 and Q6. This is comparable to the results reported by [24], where participants from the construction team interviewed were asked to rank seven performance characteristics of the Yelwa village bottle houses on a scale of one to five. It is important to note that participants ranked the high strength and stability of the bottle bricks first and ranked the good interior temperature of the bottle house as second compared to the other characteristics. Additionally, the project coordinator for the Yelwa bottle house further reports that the plastic bottle house was able to maintain the interior at a constant temperature of 18 °C, which is beneficial for hot climates [46].

*4.2. Predicted Mean Votes (PMV)*

Fanger's theory [47] was used to calculate the PMV values using the measured parameters. The PMV and PPD values were calculated using the testing equipment (Testo 480 multifunction meter). The PPD is calculated based on the number of thermally dissatisfied people in a group using the predicted mean vote (PMV). The operative temperature for each dwelling was calculated as the average of the air temperature and the mean radiant temperature according to ASHRAE Standard 55 [31]. According to ASHRAE standard 55 (2010), the operative temperature can be calculated using the formula below where

occupants are sedentary, having metabolic rates between 1.0 and 1.3 met, are not in direct sunlight, and are not exposed to air velocities greater than 0.20 m/s:

$$T_o = (T_a + T_r)/2 \qquad (1)$$

where $T_o$ = operative temperature, $T_a$ = air temperature, and $T_r$ = mean radiant temperature.

From Table 7, it can be observed that the average air temperature for the bottle house over the measurement period was 32.2 °C, and the mean radiant temperature was 32.7 °C, with an operative temperature of 32.45 °C. These values are much lower than those recorded in the other dwellings. This could be attributed to the overall components of the building envelope in the bottle house having higher thermal resistance compared to that of the other dwellings [21]. This improved thermal performance is also evident in the PMV, with the bottle house also recording a lower percentage of persons dissatisfied (PPD) than the other dwellings. The results obtained were similar to those from Adaji et al. [25], who conducted a similar thermal comfort survey in Abuja, Nigeria, and recorded average indoor temperatures between 30.0 °C and 31.7 °C for concrete masonry houses. Additionally, the maximum indoor temperature values of 32.6 °C for the bottle house used in this study is much lower than those observed by Mokhtar et al. [23] in their plastic bottle eco-house, although the minimum value is higher in comparison.

**Table 7.** Measured temperature and PMV values.

| House Type | Values | MRT (°C) | Air Temp (°C) | Operative Temp (°C) | RH (%rH) | Air Velocity (m/s) | PMV Calc | PPD Calc (%) |
|---|---|---|---|---|---|---|---|---|
| Bottle House | Mean | 32.7 | 32.2 | 32.5 | 54.0 | 0.02 | 2.7 | 96.2 |
|  | Min | 32.1 | 31.9 | 32.0 | 51.0 | 0.01 | 2.6 | 94.7 |
|  | Max | 34.9 | 32.6 | 33.8 | 56.1 | 0.05 | 3.0 | 99.1 |
| Mud House 1 | Mean | 35.0 | 34.7 | 34.9 | 51.1 | 0.05 | 3.0 | 99.1 |
|  | Min | 35.0 | 34.6 | 34.8 | 49.9 | 0.03 | 3.0 | 99.1 |
|  | Max | 35.0 | 34.8 | 34.9 | 51.7 | 0.06 | 3.0 | 99.1 |
| Mud House 2 | Mean | 39.0 | 36.8 | 37.9 | 47.5 | 0.02 | 3.0 | 99.1 |
|  | Min | 36.6 | 36.0 | 36.3 | 42.2 | 0.02 | 3.0 | 99.1 |
|  | Max | 43.4 | 38.2 | 40.8 | 55.0 | 0.04 | 3.0 | 99.1 |
| Cement House 1 | Mean | 33.7 | 33.2 | 33.45 | 60.4 | 0.02 | 3.0 | 99.1 |
|  | Min | 33.5 | 32.8 | 33.15 | 59.5 | 0.01 | 3.0 | 99.1 |
|  | Max | 34.4 | 33.5 | 33.95 | 61.0 | 0.03 | 3.0 | 99.1 |
| Cement House 2 | Mean | 35.3 | 34.4 | 34.9 | 58.0 | 0.06 | 3.0 | 99.1 |
|  | Min | 34.6 | 34.3 | 34.5 | 55.8 | 0.03 | 3.0 | 99.1 |
|  | Max | 36.6 | 34.6 | 35.6 | 58.8 | 0.08 | 3.0 | 99.1 |

Acceptable thermal sensations, according to the PMV, model should typically fall between −1 and +1 on the scale [48]; therefore, these values indicate that the occupants are uncomfortable as they fall between +2 and +3 on the PMV scale. However, the PMV model has been tested over the years and has been noted to better predict thermal sensations in air-conditioned buildings when compared with non-air-conditioned buildings, as it tends to overestimate the feeling of warmth [32]. This is supported by the fact that the thermal sensation votes (TSVs) from the occupants are different from the PMV results recorded. In this study, 50% of the participants voted that they found their thermal environment comfortable, as seen in Table 3. All three occupants of the bottle house voted −2 on the TSV scale (Table 6), indicating that they felt cool within the dwelling, which, according to two out of the three participants, was comfortable, while for one participant, this was uncomfortable. However, the PMV calculated for these occupants ranged between +2.6 and 3.0 (very warm–hot). Similarly, in the other dwellings, the PMV calculated was +3 for

all occupants, indicating that the indoor environment was hot and uncomfortable. The TSV results, in contrast, show that the occupants voted between 0 and +3, which signifies that they felt slightly warm to hot, except for one person, who felt neutral. Only 4 out of 16 participants' true thermal sensations were accurately predicted via the PMV model. The difference is particularly glaring when comparing the thermal sensation votes for the occupants in the bottle house to the calculated PMV. This result can be compared to previous research in similar climates, which shows that the PMV comfort model overestimates the thermal comfort sensation of building occupants [49–51]. Hamzah et al. [50] carried out a survey of eight secondary schools in Indonesia using questionnaires to collect data to determine the thermal comfort of the students based on TSV. Although they recorded high air and radiant temperatures ranging from 28.2 to 33.6 °C, 80% of the students surveyed reported that they were comfortable. Furthermore, previous studies have reported that occupants in naturally ventilated buildings in warm climates typically experience a higher neutral temperature. A study by Efeoma and Uduku [52] in hot–humid Nigeria reports an acceptable comfort range of 25.4–32.2 °C and a neutral temperature of 28.8 °C. Similarly, Adaji et al. [25] report a neutral temperature of 28.0–30.4 °C for Abuja. They further explain that due to exposure to high temperatures above 28 °C, some of the occupants have adapted to their thermal environment, hence the high neutral temperatures recorded.

The errors resulting from this overestimation of PMV, however, can be combated by using the extended PMV model for "non-air-conditioned buildings in warm climates" developed by Fanger and Toftum [32], which considers the different expectations of building occupants. This is calculated by using an expectancy factor, e, for the region under study, in this case, Abuja, Nigeria. The expectancy factor is used to multiply the recorded PMV and yield a better estimate of the thermal sensations of the building occupants in naturally ventilated buildings. According to Fanger and Toftum [32], for regions with year-round warm weather with no or few air-conditioned buildings, an expectancy factor of 0.5 or 0.7, respectively, should be used. This study used the expectancy factor of 0.7 to obtain the adjusted PMV in Table 8.

**Table 8.** Comparison of observed thermal sensation votes (TSVs) with the new PMV model.

|  | Expectancy Factor, e | Mean PMV Recorded | PMV Adjusted to Occupants' Expectation | Mean Thermal Sensation Votes |
|---|---|---|---|---|
| BH | 0.7 | 2.7 | 1.9 | −2.0 |
| MH 1 | 0.7 | 3.0 | 2.1 | 2.0 |
| MH 2 | 0.7 | 3.0 | 2.1 | 2.0 |
| CH 1 | 0.7 | 3.0 | 2.1 | 2.3 |
| CH 2 | 0.7 | 3.0 | 2.1 | 2.5 |

Using the extended PMV model brings the PMV values much closer to the TSV values, as can be seen in Table 8 above. On the PMV scale, +2.1 is considered 'Warm', which is a more accurate thermal sensation than 'Hot' (+3), which was initially recorded for the other dwellings. Furthermore, Fang et al. [53] observed in their study that with operative temperatures higher than 34 °C, PMV is less accurate, and a discrepancy between PMV and TSV arises with increasing operative temperature. It is noteworthy to mention that although the PMV adjusted to occupants' expectations for the other dwellings are similar to the mean thermal sensation votes, that for the bottle house is different and is still significantly less than that recorded for the other dwellings. The slightly better performance could be attributed to the deliberate design features incorporated to improve thermal performance, as detailed in [7]. These include water-filled bottles to increase thermal mass, orientation of the building to improve natural ventilation, and light-colored painted walls to reduce radiant temperature.

Comparing the bottle house (BH) to the cement-based house (CH), the construction of the bottle house costs only 35% of the total cost for a one-bedroom bungalow in Nigeria [7].

By providing an alternative for affordable housing through upcycling of plastic waste, this study provides a solution to the current housing deficits of 28 million dwellings in Nigeria [54], which is in line with the UN SDG 11 'to foster sustainable communities' [2]. In addition, urbanization and economic growth are leading to increased energy demand, especially in developing countries. Therefore, adopting this BH prototype for affordable housing improves occupant thermal comfort, as is evidenced in this study, which will result in less reliance on cooling and invariably less operational energy consumption by these occupants. Furthermore, the National Bureau of Statistics NBS (2018) reports that, on average, only 63.7% of Nigerians have access to electricity, and this figure is 69.5% in Abuja. Moreover, they report that electricity from the national grid is only available on average for 6–8 h a day, with people having to rely on off-grid power generators for up to 4.1 h a day. According to Märzinger and Österreicher [55], buildings account for up to one-third of worldwide energy consumption; thus, it is important for buildings to provide thermal comfort for inhabitants while operating efficiently. Furthermore, Gong et al. [56] emphasize the need to minimize energy usage while improving occupant thermal comfort.

### 4.3. Results of the Simulation Study

Due to the high values recorded for the other buildings used in this study, the authors have made further recommendations to improve the thermal comfort of occupants in the cement and mud houses. This will take the form of retrofit interventions to further reduce the heat gains to the indoor spaces. The use of the EnergyPlus simulation interface in DesignBuilder was crucial to quantifying the effects of these retrofit interventions for the buildings.

One of the cement houses was selected from the case study buildings to serve in quantifying interventions needed to further improve the buildings to a more comfortable temperature similar to the bottle house.

Figure 7 shows the exterior and interior of the selected building. Shading for the windows comes from the overhanging roof and the tinted glazing of the windows. Additionally, curtains were used indoors for privacy. The windows were also fitted with mosquito netting and security bars. Electric fans were observed inside the living spaces, which demonstrates the need for additional cooling to improve thermal comfort. The floor is finished with ceramic/porcelain tiles, and the walls are painted only indoors. When asked during the interviews, the homeowner mentioned that it was common to leave the external wall face unpainted in their community. The roof was finished with dark-colored Aluzinc roofing sheets, which could increase heat gains into the building when compared with bright colors, which are more reflective and recommended for hot climates [57].

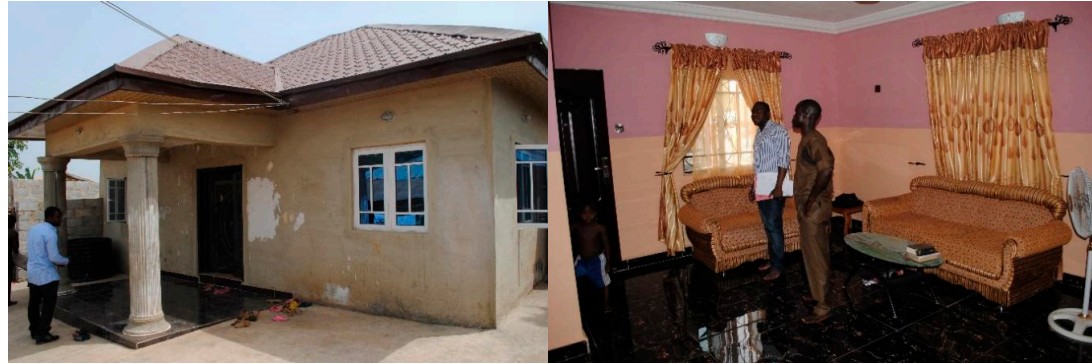

**Figure 7.** Building exterior and interior.

Figure 8 shows the building geometry replicated in DesignBuilder with the building specifications input into the relevant templates for the EnergyPlus simulations. This represents the baseline model.

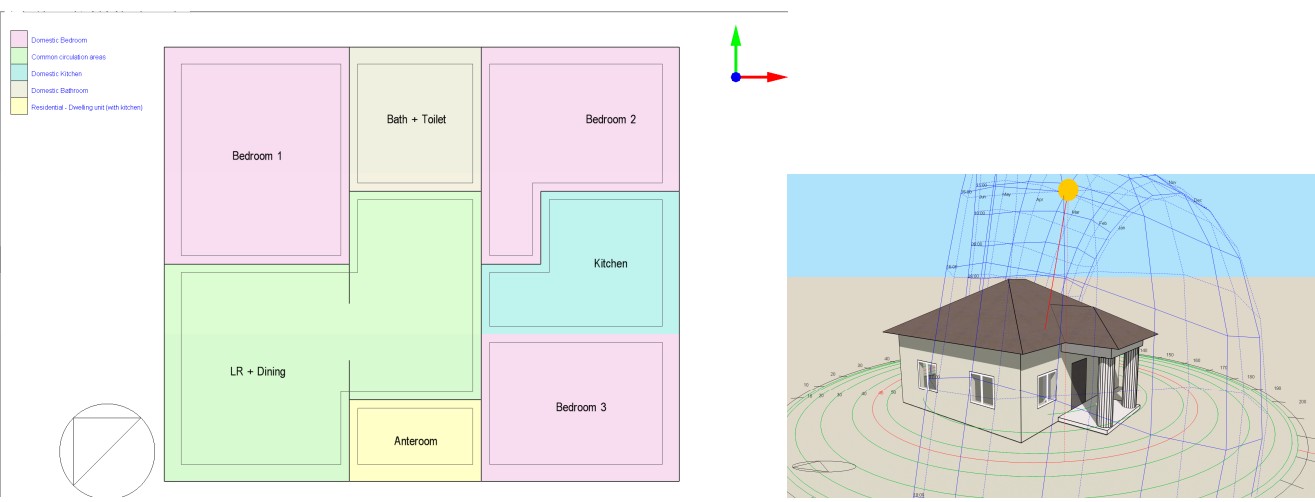

**Figure 8.** DesignBuilder model of selected building.

Monthly averages for indoor operative temperature range from 28.34 °C to 31.10 °C, with relative humidity ranging from 47.71% to 88.81% during the dry and rainy seasons, respectively (Figure 9). Using measured outdoor temperature data from the site for cooling design simulations, the sub-hourly indoor temperature values can be seen in Figure 10. This was carried out to represent a typical day (10th of April) in one of the hottest months in Nigeria. It also coincides with the timeline during which the quantitative and qualitative thermal measurements were conducted for this study. Indoor operative temperature values range from 29.29 °C to 35.82 °C throughout the day. These values are quite similar to the measured values for the cement houses shown in Table 7. However, they exceed the neutral temperature of 28 °C to 30.4 °C reported by [25] for occupants in residential buildings in Abuja. The highest heat gains into the building zones were observed through the walls, general lighting, roof/ceiling and then the glazing during the peak periods of the day. The simulation outputs have been summarized below in Table 9.

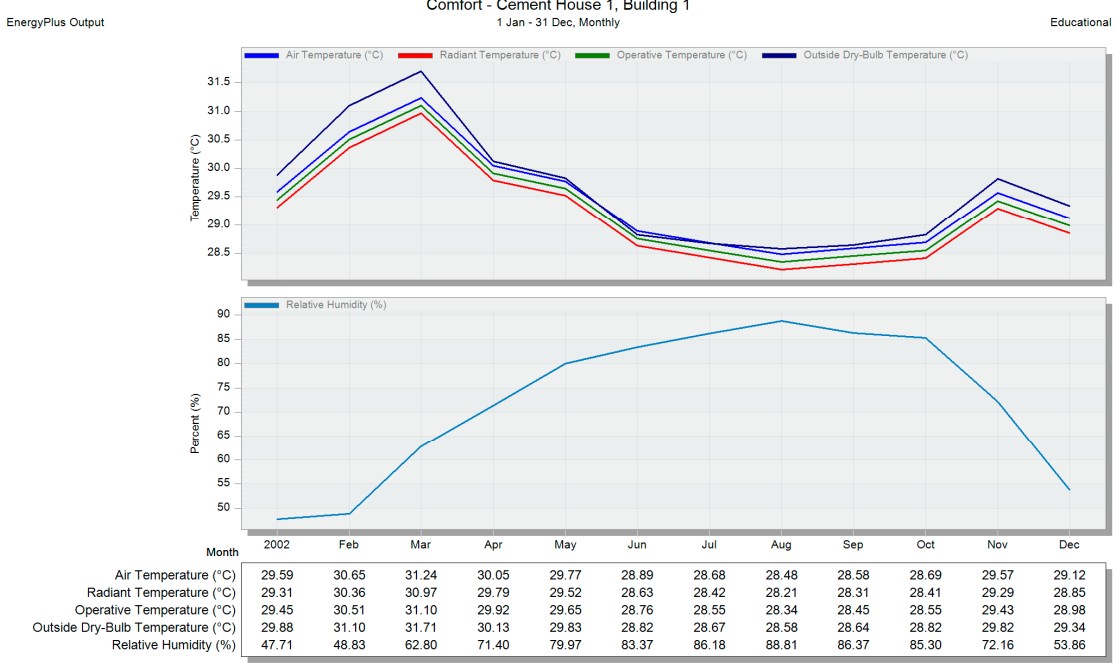

**Figure 9.** Indoor operative temperature values for all months showing highest values between February and April.

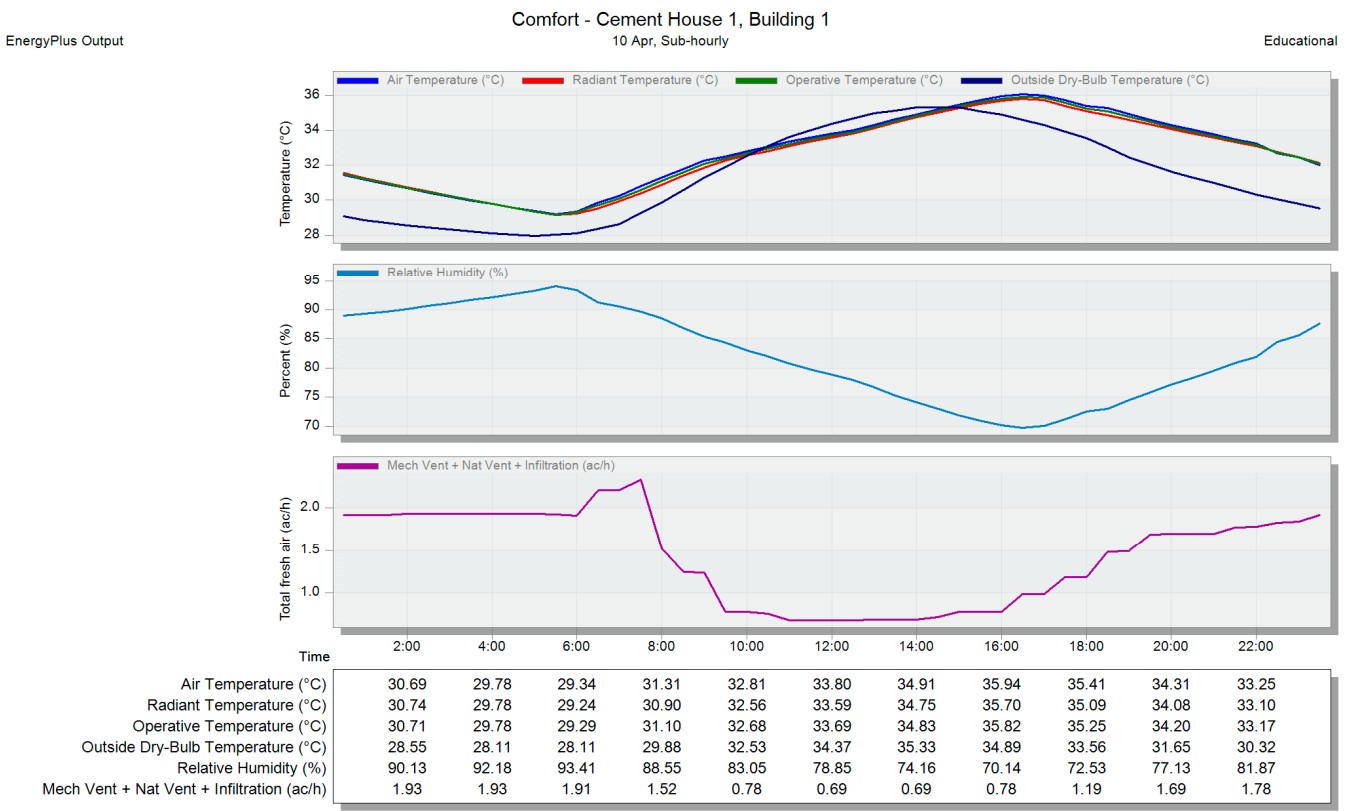

**Figure 10.** Cooling design for a typical day of one of the hottest months showing indoor operative temperature.

**Table 9.** EnergyPlus simulation output for the baseline model.

| Baseline | January | February | March | April | May | June | July | August | September | October | November | December |
|---|---|---|---|---|---|---|---|---|---|---|---|---|
| Ext. Walls (kWh) | 546.55 | 572.72 | 674.14 | 558.85 | 575.91 | 491.67 | 490.25 | 476.41 | 464.53 | 488.71 | 538.80 | 517.66 |
| General lighting (kWh) | 282.91 | 255.21 | 281.20 | 270.81 | 273.87 | 259.60 | 271.37 | 274.85 | 269.06 | 282.45 | 273.06 | 280.91 |
| Roof/ceilings (kWh) | 231.85 | 240.44 | 285.49 | 233.58 | 225.45 | 191.43 | 192.14 | 179.88 | 179.00 | 190.63 | 216.43 | 208.68 |
| Solar gains exterior window (kWh) | 189.16 | 176.03 | 186.23 | 188.80 | 186.47 | 178.08 | 181.31 | 176.79 | 173.92 | 172.69 | 176.90 | 187.55 |
| Fuel total (electricity, kWh) | 898.01 | 933.77 | 1163.70 | 1021.06 | 1044.94 | 894.64 | 913.48 | 895.88 | 875.60 | 914.58 | 938.34 | 851.54 |
| Carbon emissions (kg) | 544.19 | 565.86 | 705.20 | 618.76 | 633.23 | 542.15 | 553.57 | 542.91 | 530.61 | 554.23 | 568.63 | 516.04 |

### 4.4. Interventions Considered and Results

Although the highest u-values were from the building walls, the authors have not recommended the use of insulation for the walls due to the large wall area. It could cause heat to be trapped indoors, especially in hot–humid climates, and due to the high levels of humidity, and could lead to mold growth and deterioration of the walls. In addition, the use of external wall cladding was not considered, as this would incur more costs for the building owners, and the weight may not be accounted for in the building structure. Therefore, since the house is situated within a low-income community, the interventions recommended have to be effective, easily achievable, and inexpensive. Hence, the authors considered the use of roof insulation, which has been recommended by previous authors [58,59], as an effective solution to curb solar heat gains in buildings situated along the equator (Figures 11 and 12). In addition, the authors considered the replacement of incandescent bulbs with LED bulbs to reduce internal heat gains (Figures 13 and 14). The simulation outputs have been summarized below in Table 10.

Through the use of dynamic thermal modeling, these interventions demonstrate improvements to the indoor operative temperature, reducing energy consumption and, in turn, operational carbon emissions. Using 100 mm EPS as roof insulation for the building

resulted in a 1.1% reduction in operative temperature, a 45.3% reduction in heat gains through the roof, and a 4% reduction in operational energy and carbon emissions for all months. Furthermore, changing the lighting from incandescent to LED bulbs resulted in an overall reduction of 2.9% in operative temperature, an 89.1% reduction in internal heat gains from lighting, and a 36.9% reduction in operational energy and carbon emissions annually. The range of monthly indoor operative temperature improved from 28.34 °C–31.10 °C to 27.56 °C–30.17 °C. This falls within the neutral temperature reported for occupants in Abuja, Nigeria [23]. However, for the cooling design simulation, which was used to show values of a typical summer day using the measured outdoor temperatures, the range of values changed from 29.29 °C–35.82 °C to 28.62 °C–34.06 °C. This is still higher than the neutral temperature range reported for the study context, although the number of hours the building is overheating and the temperature values have drastically reduced.

It is noteworthy to mention that during the interviews, one of the residents in the cement house mentioned how they had not had electricity from the national grid for about 4 to 6 months and had to rely on off-grid power generators, which imposes an additional cost on these homeowners. These retrofit interventions could, therefore, provide a pathway to improving their indoor environment.

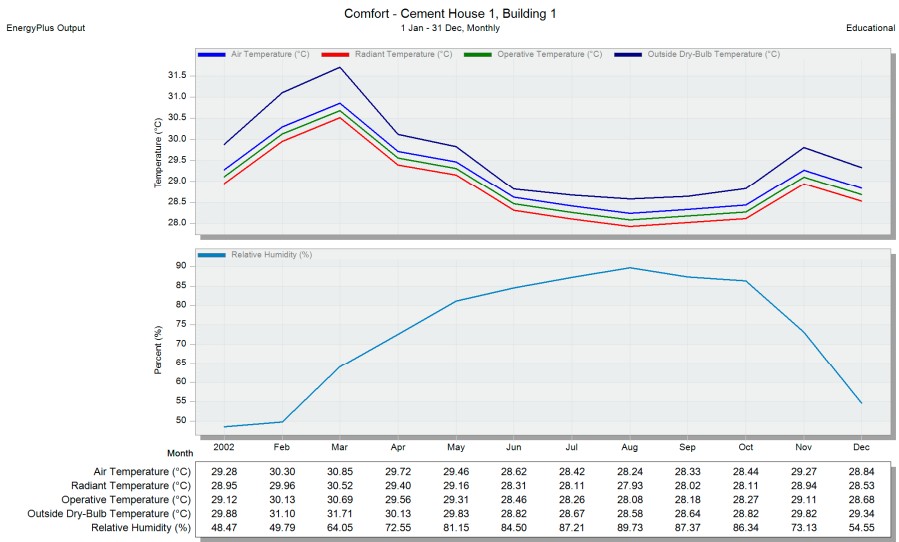

**Figure 11.** Improvements to indoor operative temperature values from the addition of roof insulation.

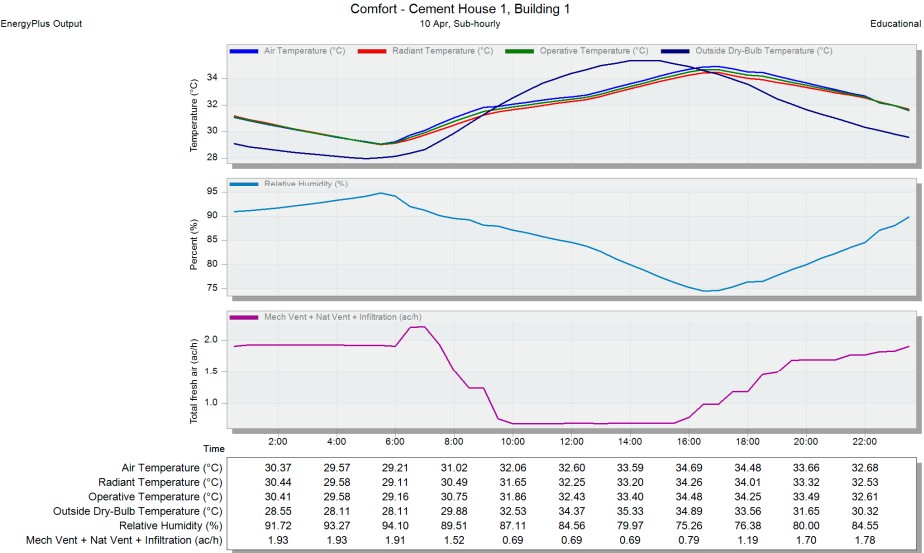

**Figure 12.** Sub-hourly values for indoor operative temperature from the addition of roof insulation.

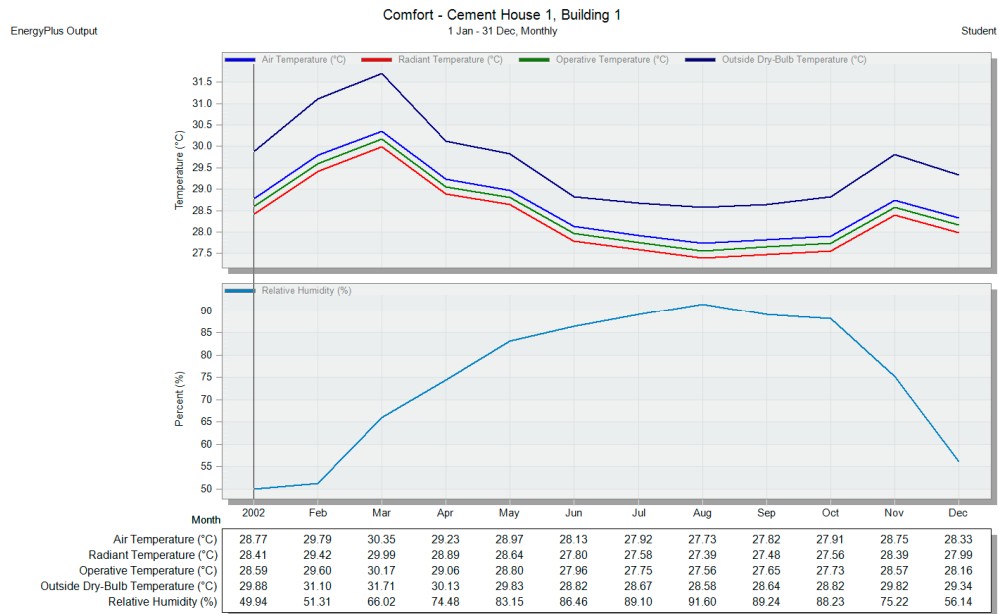

**Figure 13.** Improvements to indoor operative temperature values from the addition of roof insulation and the use of energy-efficient lighting.

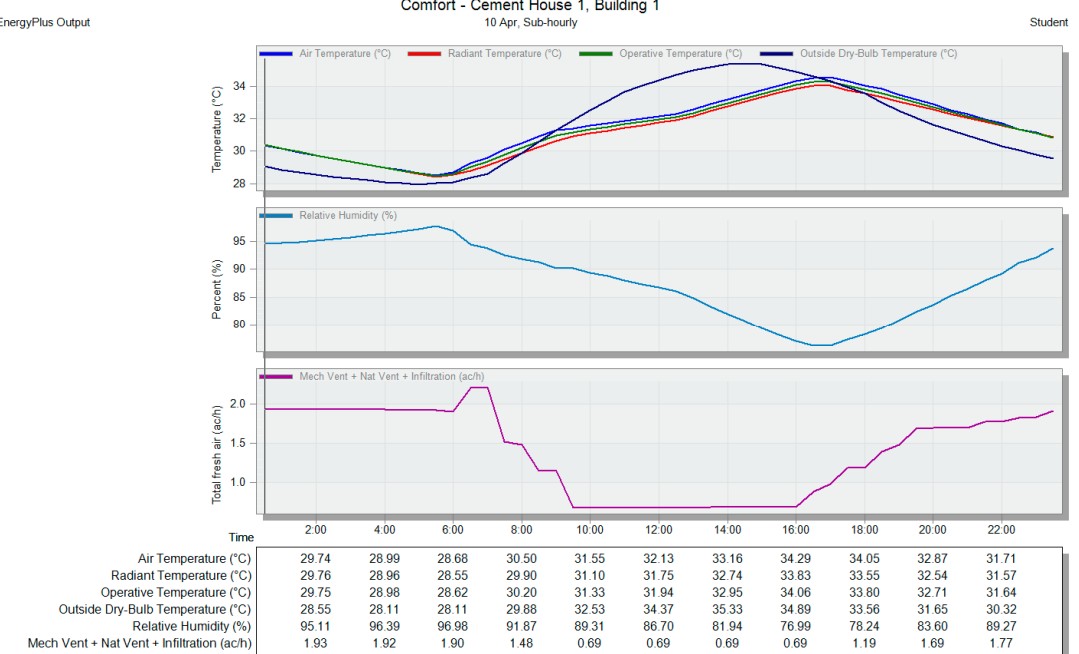

**Figure 14.** Sub-hourly values for indoor operative temperature from the addition of roof insulation and use of energy-efficient lighting.

**Table 10.** EnergyPlus simulation output for the building with 100 EPS roof insulation and LED lighting.

| Roof Insulation + LED | January | February | March | April | May | June | July | August | September | October | November | December |
|---|---|---|---|---|---|---|---|---|---|---|---|---|
| General lighting (kWh) | 30.33 | 27.39 | 30.38 | 29.36 | 30.27 | 29.42 | 30.27 | 30.33 | 29.36 | 30.27 | 29.36 | 30.43 |
| Roof/ceilings (kWh) | 127.24 | 128.89 | 148.52 | 119.73 | 118.99 | 102.65 | 106.45 | 104.25 | 101.61 | 109.04 | 119.04 | 117.24 |
| Fuel total (electricity, kWh) | 544.17 | 604.30 | 792.34 | 671.80 | 690.40 | 565.54 | 570.00 | 550.44 | 537.20 | 559.01 | 592.02 | 505.19 |
| Carbon emissions (kg) | 329.77 | 366.20 | 480.16 | 407.11 | 418.38 | 342.72 | 345.42 | 333.57 | 325.54 | 338.76 | 358.77 | 306.15 |

## 5. Conclusions

This study has considered the use of waste plastic bottles for housing construction for low-income householders in Nigeria, which is novel when compared to existing research. The study went further to conduct post-occupancy surveys for the bottle house to determine the thermal sensation votes (TSVs) of the occupants and compare this with the predicted mean vote (PMV) calculated using experimental readings. These figures were also compared to two popular building typologies (mud and cement), which are typically found in this location. The results of this paper show that the occupants of the bottle house felt thermally comfortable even though measured indoor conditions suggest otherwise. In any case, the bottle house had the best performance compared to the other houses. This was attributed to the measures incorporated during the construction of the bottle house. Furthermore, the use of a simulation study helped proffer solutions to further improve the indoor temperature for the buildings used in this study. The addition of roof insulation and changing the lighting bulbs from incandescent bulbs to LEDs resulted in an overall reduction of 2.9% in operative temperature, an 89.1% reduction in internal heat gains from lighting, a 45.3% reduction in heat gains through the roof, and a 36.9% reduction in operational energy and carbon emissions annually.

Although this paper provides novel insights into the in situ thermal performance of the bottle house, it has a few limitations. One such limitation is the sample size, as there were only 16 respondents in total representing the five families interviewed. Additionally, compared to the other housing types evaluated, which are more common, the plastic bottle house is the only prototype within this community, and few other plastic bottle house studies have been undertaken for comparison. Therefore, future studies will consider a larger sample size, where possible, and also include a longer period of thermal comfort survey and environmental monitoring over different time periods of the year, not just during the hottest month, as used in this study. The TSV records the short-term thermal perception of occupants; as such, longer periods of observation and measurement will immensely improve the findings of this study. Future research will address the limitations highlighted. Additionally, with more iterations of the bottle house prototype within the study context and in similar climes, there would be more data to improve comparison.

**Author Contributions:** Conceptualization, N.C.O. and M.A.O.; Data curation, A.B. and M.A.O.; Formal analysis, N.C.O.; Funding acquisition, A.B. and M.A.O.; Investigation, N.C.O., A.B. and M.A.O.; Methodology, A.B. and M.A.O.; Project administration, A.B. and M.A.O.; Resources, A.B. and M.A.O.; Writing—original draft, N.C.O., A.B., O.F.A. and M.A.O.; Writing—review and editing, N.C.O., A.B., O.F.A. and M.A.O. All authors have read and agreed to the published version of the manuscript.

**Funding:** This research was funded by the Royal Academy of Engineering [grant number FoESF1617\1\13] through the Frontiers of Engineering for Development seed corn funding 2016.

**Institutional Review Board Statement:** The study was conducted in accordance with the Ethics Guidelines of DE MONTFORT UNIVERSITY and approved by the Computing Engineering and Media Ethics Committee (March 2019). Consent was obtained from participants before proceeding with data collection.

**Informed Consent Statement:** Informed consent was obtained from all subjects involved in the study.

**Data Availability Statement:** Data will be made available upon request.

**Conflicts of Interest:** The authors declare that they have no known competing financial interests or personal relationships that could have appeared to influence the work reported in this paper. The funders had no role in the design of the study; in the collection, analyses, or interpretation of data; in the writing of the manuscript; or in the decision to publish the results.

## Appendix A

Copy of thermal comfort questionnaire.

## Study of Thermal Comfort in Homes and offices

## Comfort Survey-01 (0 min from start of survey)

**Q.1:** **Your activity level now** (Please circle around the appropriate):

Reclining Seated         relaxed Seated         Sedentary work

**Q.2:** **How are you feeling at this precise moment?**

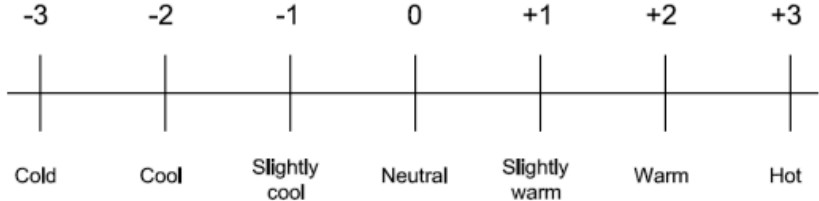

**Q.3:** **Do you find this . . . ?**

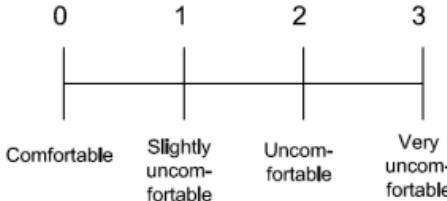

**Q.4:** **Please state how you would prefer to be now?**

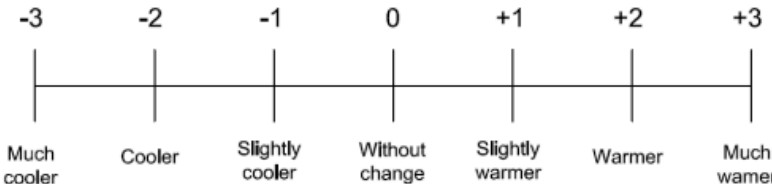

**Q.5:** **How do you judge this environment (local climate) on a personal level?** (Please circle around the appropriate):

Clearly acceptable     Just acceptable       Just unacceptable     Clearly unacceptable

**Figure A1.** *Cont.*

## Study of Thermal Comfort in Homes and offices

**Q.6:** **Please state your personal tolerance of this environment. Is it . . .?**

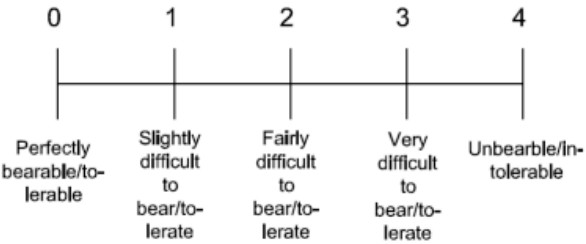

**Q.7:** **Time of completing the survey:** _______________________________

**Q.8:** **Your total clothing insulation level:** _______________________________

(Please use the clothing insulation list given at the back of this pack to calculate the total value of your clothing insulation)

**Q.9:** **Please describe any changes you have made to your clothing, your environment or to yourself**

(i.e. Changes clothing, open/close window, open/close door, drinking (hot or cold)/eating)

_______________________________________________________________

*Thank you for finishing this stage of the survey. Please do remember to complete the remaining surveys at every 15 minutes interval.*

**Figure A1.** Copy of questionnaire used for qualitative data collection.

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
