# Peer review of "The Bottle House: Upcycling Plastic Bottles to Improve the Thermal Performance of Low-Cost Homes"

_sustainability, doi:10.3390/su16041360_

Round 1

Reviewer 1 Report

Comments and Suggestions for Authors

They mention that they provide novel information about thermal performance, however, according to the results and conclusions, it is not perceived that there is a contribution or novelty.

To a large extent it is documented that the bottles do not contribute much to thermal comfort, especially because they are placed on the walls. The geometric parameters of the height of a home contribute more to the thermal comfort, in addition, the vegetation and constructions around the homes significantly influence the thermal comfort. Furthermore, it may not be appropriate to make conclusions based on a survey, without thinking about whether the survey questions are appropriate.

Comments on the Quality of English Language

Review the wording.

Author Response

Dear Reviewer,

We thank you for your constructive comments which have served to further enrich this manuscript. Please find the responses to the comments attached.

Kind regards!

Reviewer 2 Report

Comments and Suggestions for Authors

Journal: SUSTAINABILITY
Manuscript ID :
sustainability-2753617
Title : "
Bottle House: Upcycling Plastic Bottles to Improve the Thermal 2 Performance of Low-Cost Homes"
Author(s):

Nwakaego C. Onyenokporo, Arash Beizaee2, Olutola F. Adekeye, and Muyiwa A. Oyinlola

General remark:

In my opinion, the paper " Bottle House: Upcycling Plastic Bottles to Improve the Thermal 2 Performance of Low-Cost Homes” was carefully written by the authors.  For this  reason, I consider the paper may be accepted after minor corrections, and published in SUSTAINABILITY

Comment 1: Introduction

Please briefly explain the background well enough that researchers outside your specialty can understand it. Do you consider the topic original or relevant in the field? Does it  address a specific gap in the field?

Please give the reasons for performing this study.

Comment 3: Methodology

Please It would be necessary to make an ANOVA to analyze all your data? On the other hand, it is important to explain the experimental design you used.

Comment 4: literature

Please compare your results to previous literature. Please give the specific improvements of your work in comparison to previous

Comment 5: conclusions

Please you should interpret the results, place them in context of previous findings.

Comment 6: references

Please the use of DOI is highly recommended.

Author Response

(The authors gave the same response as above.)

Reviewer 3 Report

Comments and Suggestions for Authors

1. Some grammar errors are found in line 22 ( change to were). 

2. Similarly, line 23. Do not repeat the words. Use only acronym.

3. Use metric tons in line 43, 45 46 etc. 

4. Line 48. Start like this. However, these infrastructure......

5. Similarly, line 78 and 79 need comas at appropriate places.

Novelty:

1.       I do not see novelty as described by the authors. The authors claim that a knowledge gap is there for the qualitative aspect of in-situ building. However, their research is mainly with Nigeria and similar studies have been conducted in other locations previously, as mentioned in the manuscript, by other authors. The authors also pointed out that the previous report have been available for thermal performance of bottled houses. Thermal performance is the main issue of the manuscript.

2.       Thermal comfort is conducted using survey. I am very skeptical that bottle houses are more comfortable than mud and cement houses. The proof is not clear, I believe.

3.       The authors claim the non-existing discovery of comfort level of bottle house, which needs more research supported data. I believe the claim itself is wrong.

4.       However, I believe the authors have quoted relevant citations/references from the previous published articles.

Final comment: I think authors need to rephrase the structure and relevancy of this manuscript with major revision.

Comments on the Quality of English Language

English language needs to be improved in certain areas.

Author Response

(The authors gave the same response as above.)

Round 2

Reviewer 3 Report

Comments and Suggestions for Authors

The authors changed the perspective of the paper by adding questionaire and improved the content of the paper.